# Precancerous liver diseases do not cause increased mutagenesis in liver stem cells

Luan Nguyen [1,5], Myrthe Jager [1,5], Ruby Lieshout [2], Petra E. de Ruiter[2], Mauro D. Locati[1], Nicolle Besselink [1], Bastiaan van der Roest [1], Roel Janssen [1], Sander Boymans [1], Jeroen de Jonge[2], Jan N. M. IJzermans [2], Michail Doukas [2], Monique M. A. Verstegen[2], Ruben van Boxtel [3], Luc J. W. van der Laan [2], Edwin Cuppen [1,4 ✉] & Ewart Kuijk [1✉]

Inflammatory liver disease increases the risk of developing primary liver cancer. The mechanism through which liver disease induces tumorigenesis remains unclear, but is thought to occur via increased mutagenesis. Here, we performed whole-genome sequencing on clonally expanded single liver stem cells cultured as intrahepatic cholangiocyte organoids (ICOs) from patients with alcoholic cirrhosis, non-alcoholic steatohepatitis (NASH), and primary sclerosing cholangitis (PSC). Surprisingly, we find that these precancerous liver disease conditions do not result in a detectable increased accumulation of mutations, nor altered mutation types in individual liver stem cells. This finding contrasts with the mutational load and typical mutational signatures reported for liver tumors, and argues against the hypothesis that liver disease drives tumorigenesis via a direct mechanism of induced mutagenesis. Disease conditions in the liver may thus act through indirect mechanisms to drive the transition from healthy to cancerous cells, such as changes to the microenvironment that favor the outgrowth of precancerous cells.

[1] Center for Molecular Medicine and Oncode Institute, University Medical Center Utrecht, Utrecht, The Netherlands. [2] Erasmus Medical Center, Rotterdam, The Netherlands. [3] Princess Máxima Center, Utrecht, The Netherlands. [4] Hartwig Medical Foundation, Amsterdam, The Netherlands. [5] These authors contributed equally: Luan Nguyen, Myrthe Jager. ✉email: E.P.J.G.Cuppen@umcutrecht.nl; e.w.kuijk-3@umcutrecht.nl

Liver cancer is the fifth most common cancer worldwide, causing around 720,000 deaths each year[1]. Different sub-types of primary liver cancer can be recognized, of which hepatocellular carcinoma (HCC; originating from hepatocytes) and intrahepatic cholangiocarcinoma (CCA; originating from cholangiocytes) form the largest groups, together constituting over 85% of all primary liver cancers[2]. Several factors have been linked to increased HCC risk, including chronic alcohol consumption[3], as well as metabolic associated fatty liver disease (MAFLD), and its more progressive form nonalcoholic steato-hepatitis (NASH), which can be caused by obesity, diabetes, drugs/medication and metabolic conditions[4]. These factors have also been linked to an increased risk for intrahepatic CCA[5]. Though less common, chronic inflammation and fibrosis of the biliary tracts, known as primary sclerosing cholangitis (PSC), also confers increased risk of developing both HCC and CCA[6]. Our knowledge on how these environmental conditions drive liver cancer is still incomplete[7].

Chronic alcohol consumption is thought to enhance the mutational load through the metabolite acetaldehyde, which has been reported to be directly mutagenic[8] and indirectly through the formation of reactive oxygen species[9–14]. Increased burden of somatic mutations has also been observed in non-alcoholic liver disease[15]. NASH and PSC are both characterized by chronic inflammation[16,17], which may cause the production of reactive oxygen and nitrogen species that subsequently induce DNA damage[18].

However, accurate measurements of in vivo induced mutations are required to confirm that accelerated mutagenesis underlies liver tumorigenesis. We previously established a sensitive method to accurately determine all somatic mutations that have been acquired throughout life in individual human adult stem cells of the liver and gastrointestinal tract[19]. We used these catalogs of somatically acquired mutations to perform mutational signature analysis, a powerful computational method for identifying mutational processes that have been active in the life history of cells[20]. Currently, 60 single base substitution (SBS) signatures, 11 double base substitution (DBS) signatures, 18 indel signatures, and 16 structural variation (SV) signatures have been identified[20,21]. These signatures are a result of endogenous mutational sources (such as apolipoprotein B mRNA editing enzyme, catalytic polypeptide-like (APOBEC) activity[22] or homologous recombination deficiency[23,24]), but also microbial impact[25], oxidative stress[26], or anti-cancer therapies[27,28]. Mutational signature analyses on healthy human stem cells revealed that mutational processes are tissue-specific and continuously active throughout life resulting in a linear accumulation of mutations with age[19], at least under "normal" conditions. Because of the link between liver disease and liver cancer, we hypothesized that the precancerous state of liver diseases would be reflected by increased mutation rates and accumulation of mutational patterns that are characteristic to the type of DNA damage inflicted during liver tumorigenesis.

In this study, we aim to identify the mutational processes that contribute to the precancerous state in common human liver diseases. To achieve this goal, we have studied the accumulation of mutations in individual stem cells derived from livers of patients with alcoholic cirrhosis, NASH and PSC who received a liver transplantation. Surprisingly, we find that individual stem cells from liver patients did not show increased mutational burden overall, within liver cancer associated genes, nor to specific mutational signatures when compared to liver stem cells from healthy donors. Our findings suggest that environmental conditions drive liver tumorigenesis through means other than by increasing mutagenesis.

## Results

**Mutation rates do not increase in diseased livers.** Both main liver cell types, hepatocytes and cholangiocytes, can act as liver stem cells depending on the type of tissue damage that was inflicted[29]. Cholangiocytes show a high degree of cellular plasticity during regeneration and disease and act as facultative liver stem cells during impaired hepatocyte regeneration[29,30]. Cholangiocytes can be grown as intrahepatic cholangiocyte organoids (ICO)[31] that show long-term self-renewal, differentiation, and engraftment in mouse and rat models of liver failure[32,33]. In contrast, there are no suitable protocols for the long-term culture of human hepatocytes. We have previously exploited the proliferative capacity of individual cholangiocytes to determine mutation rates in individual liver stem cells of the healthy liver[19]. We reasoned that cholangiocytes are also suitable for the study of somatic mutation accumulation as a result of the diseased liver environment, because cholangiocytes are exposed to the same environmental conditions as the other liver cell types[34].

To determine whether a diseased liver environment leads to increased somatic mutation accumulation, we performed whole-genome sequencing (WGS) on clonal ICOs derived from liver stem cells from patients with diseased livers (Fig. 1a, b and Supplementary Data 1). Our study included: (i) 14 clones from five patients with cirrhosis as a result of chronic alcohol consumption; (ii) 13 clones from five NASH patients; and (iii) eight clones from three PSC patients. Organoid establishment success rates were lower for diseased liver as compared to healthy livers and was most challenging for material obtained from PSC patients. For each patient, a reference blood or multilineage bulk tissue sample was also sequenced to distinguish germline variants from somatic variants. The somatic mutation catalogs in diseased livers were compared to the mutation catalogs from 14 clones from liver adult stem cells derived from seven healthy donors.

In total, we identified 172,650 small mutations (single/double base substitutions or indels) in healthy (35,006), post-alcoholic (49,681), NASH (43,997), and PSC (19,521) livers (as well as 24,445 from a patient with HCC). Consistent with previous observations[19], somatic mutations accumulated linearly with age in healthy liver cells, at a rate of approximately 46 SBSs and nine indels per year (Fig. 2). The rate of SBS accumulation in alcoholic, NASH, and PSC ICOs showed no significant differences to that of healthy ICOs (Z-test, $p \geq 0.41$), and similarly, the rate of indel accumulation in disease ICOs was also comparable to that of healthy ICOs (Z-test, $p \geq 0.62$). We observed a slight increase in variance in SBS accumulation in alcoholic cirrhosis versus healthy ICOs which may suggest that alcohol consumption leads to mutagenesis in some but not all patients, though the increase in variance was weak (F-test, $p = 0.032$). Likewise, we found increased variance in SBS accumulation in PSC ICOs (F-test, $p = 0.001$), though this variance is likely due to having many more ICOs originating from one patient PSC2. We also compared the accumulation of DBSs and SVs. While the number of DBSs and SVs was too low to be conclusive ($\leq 50$ DBSs and $\leq 25$ SVs; Supplementary Fig. 2), the rate of mutation accumulation overall did not increase in diseased versus healthy ICOs. Taken together, these results suggest that chronic alcohol consumption or an inflamed liver environment does not lead to increased SBS, indel, DBS or SV accumulation in liver cells.

**The mutation profile of diseased livers is similar to that of healthy livers.** The presence of genome-wide patterns of mutations (also known as mutational signatures) reflects past activity of mutational processes in cells. Previously, the mutational signatures SBS12 and SBS16 have been associated with HCC[20,35], with SBS16 also being associated with alcohol consumption[36]. Additionally, SBS2 and SBS13 (APOBEC activity) were found to be active in numerous cancer types including CCA[20]. We expected that the mutational processes in diseased liver would be similar to those in

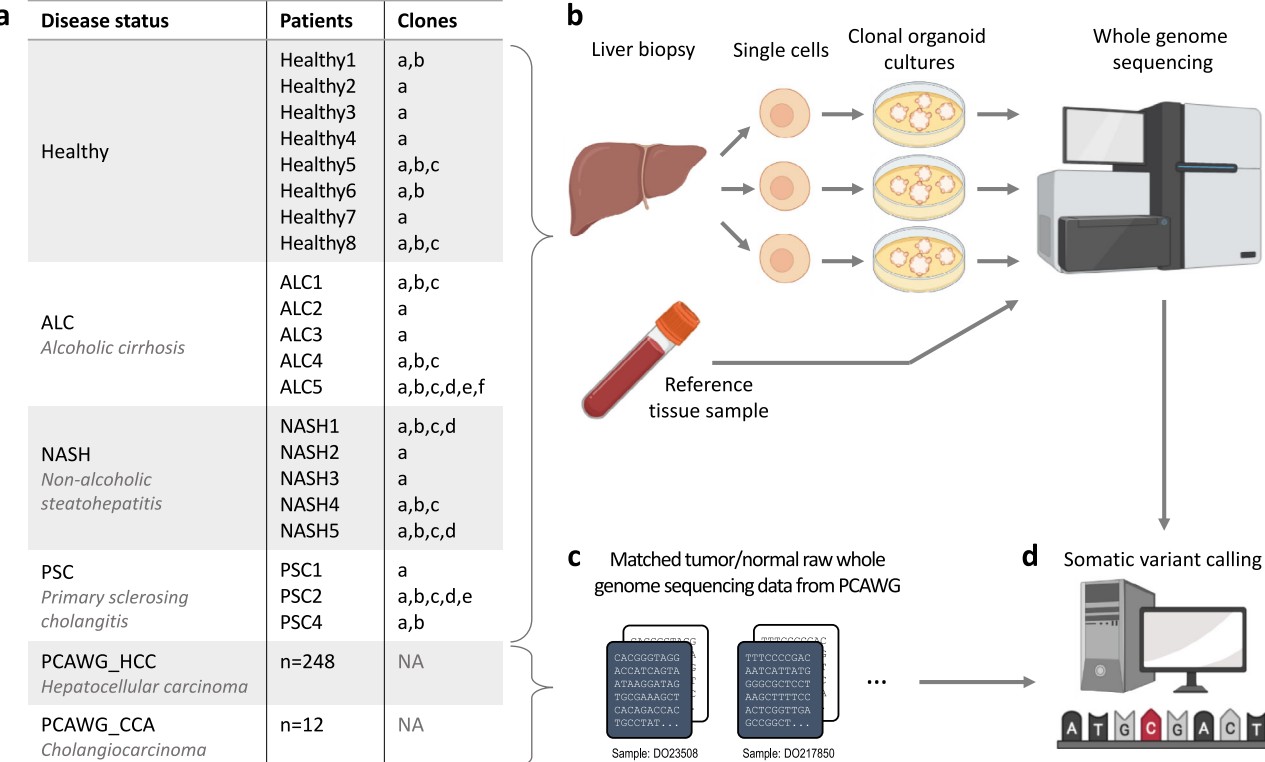

**Fig. 1 Samples and experimental setup. a** Summary of the samples used in this study. **b** Liver biopsies were taken from patients with diseased livers from which clonal intrahepatic cholangiocyte stem cell-based organoid (ICO) cultures were generated. Organoid clones were subjected to whole-genome sequencing (WGS) together with a matched tissue reference sample per patient for subtraction of germline mutations and thus detection of somatic mutations. **c** To compare the mutation profiles in liver disease versus HCC and CCA, WGS data from primary tumor samples from the Pan-Cancer Analysis of Whole Genomes (PCAWG) consortium were also included. **d** Somatic variant calling of diseased and cancerous livers was performed with the same pipeline.

cancerous liver. We thus examined whether liver disease results in increased presence of one of the above signatures or in other signatures related to liver cancer. Since the main liver cancer types are HCC and intrahepatic CCA, we selected the signatures that could be present in liver and biliary cancer based on the signature catalog from the PCAWG (Pan-Cancer Analysis of Whole Genomes) consortium[20] (see "Methods" section for further details). We ultimately quantified the presence of ten SBS and seven indel signatures in our ICOs as well as in the PCAWG HCC and CCA samples (Fig. 3). Too few DBSs and SVs were present in the diseased liver samples to perform signature analysis for these variant types (Supplementary Fig. 2).

We observed similar signature profiles in diseased and healthy ICOs, with the most predominant signatures being age-related (Fig. 3; SBS1, SBS5, SBS40, ID1, ID5, and ID8), which are present in normal cells[20]. We also found minor contributions of signatures SBS4 and ID3. These signatures are also present at a baseline level in HCC and CCA[20]. In the HCC and CCA samples from PCAWG, age-related signatures were predominant, with HCC samples also showing increased contribution of the known HCC associated signatures SBS12 and SBS16 compared to the healthy and diseased ICOs, while CCA samples showed increased contribution of APOBEC activity (SBS2, SBS13). Because these signatures were not observed in the disease ICOs, these findings argue against environmentally induced mutational processes as a force driving the transition of healthy to precancerous liver cells. It is likely that other events are required to initiate the HCC/CCA related mutational processes.

Given that certain recurrent chromosome arm gains and losses have been reported in HCC[37,38], we also examined whether liver disease leads to similar copy number alterations (CNA). We find

that aside from two samples with polyploidization (a known phenomenon in normal liver cells[39]), the genomes of the disease ICOs were relatively stable, while the genomes of HCC and CCA were clearly more unstable with known recurrent arm gains (e.g., 1q, 8q, 17q) and losses (e.g., 8p and 17p) being observed[37,38] (Supplementary Fig. 3). These data could suggest that CNA accumulation due to liver disease does not contribute to the healthy to precancerous liver transition. However, as CNAs are rare in non-cancerous cells[40], more data would be required to validate this hypothesis.

**Absence of driver gene mutations in ICOs.** Certain mutations may confer liver stem cells a growth advantage, and in diseased livers, these cells may be able to proliferate more to regenerate lost tissue. We thus examined whether the liver disease conditions resulted in positive selection of cells with non-synonymous mutations in specific genes using the *dndscv* algorithm (see details in "Methods" section). However, amongst all of the liver disease groups, no genes were found to be enriched in non-synonymous mutations ($q < 0.01$, Supplementary Data 2). In line with this result, we did not observe any coding, promoter or 5′/3′ untranslated region (UTR) mutations in driver genes of HCC and CCA (obtained from Intogen; see "Methods" section), except for one missense variant in an alcoholic cirrhosis ICO sample (*TERT* c.1588C>G) (Fig. 4). This could potentially be explained by the cells from which our disease ICOs were derived not being actual cancer precursors but only harboring passenger mutations. We acknowledge that mutations could occur in other non-coding elements but have not examined these as their impact is currently difficult to assess[41,42]. Furthermore, we also acknowledge that our small sample sizes limit our ability to find enriched driver gene

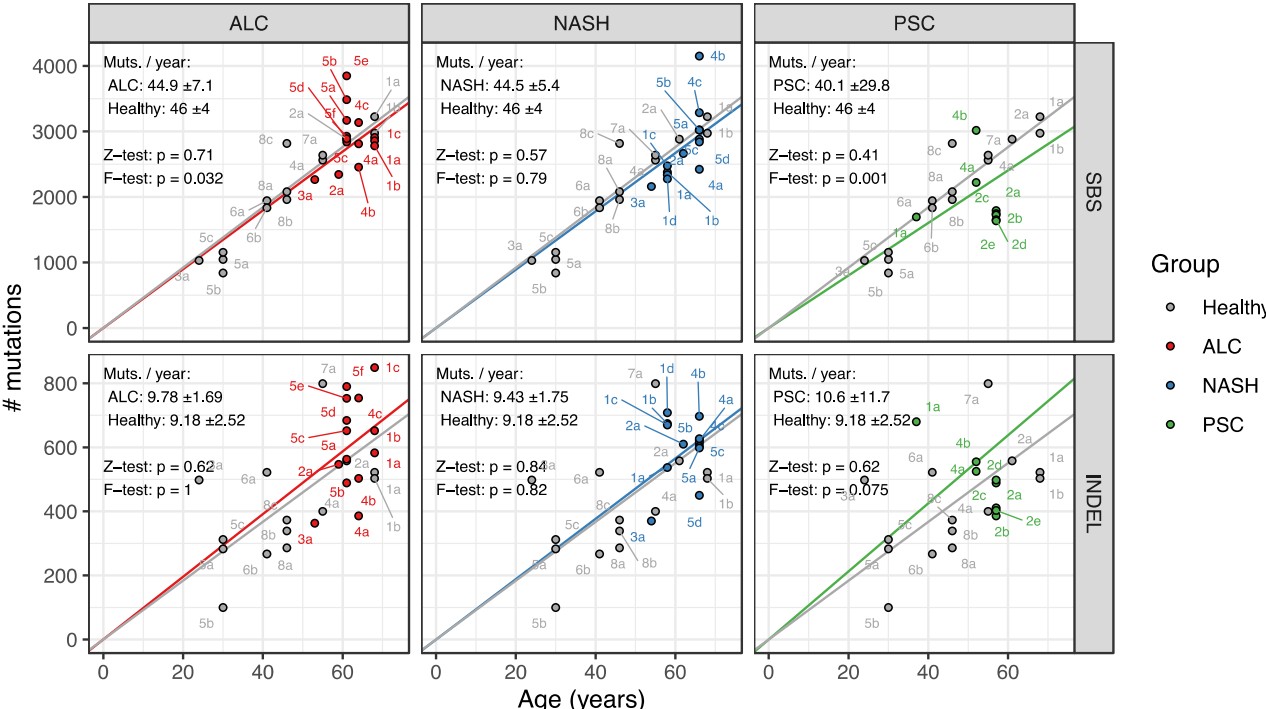

**Fig. 2 Accumulation of somatic single base substitutions (SBS) and small insertions/deletions (indel) in organoids derived from biopsies of healthy livers compared to those from patients with diseased livers.** ALC alcoholic cirrhosis, NASH non-alcoholic steatohepatitis, PSC primary sclerosing cholangitis. Each point is labelled by patient number and clone letter. Two-sided Z-tests were performed to determine whether there was a significant difference between the linear mixed effects regressions (i.e., the rate of mutation accumulation) of the disease versus healthy ICOs. One-sided F-tests were performed to determine whether there was a significant increase in variance in rate of mutation accumulation in disease samples versus healthy samples. ± values indicate the 95% confidence interval range of each regression and "p" indicates the p-values of the Z-tests and F-tests.

mutations in the diseased liver ICOs. In contrast, we found enrichment of non-synonymous mutations in *TP53* in the PCAWG CCA samples, and in *TP53* and *CTNNB1* as together with 13 other genes in the PCAWG HCC samples ($q < 0.01$, Supplementary Data 2). While *dndscv* does not consider non-coding variants, we also observed *TERT* promoter mutations in the PCAWG HCC samples ("upstream_gene_variant", Fig. 4). These genes have been reported as known cancer driver genes in the respective cancer types[43]. No mutations in these driver genes were found in 34% (84/284) of PCAWG HCC samples indicating that mutations in these genes are not necessarily a requirement for HCC development.

## Discussion

Despite the association between liver disease and primary liver cancer (which includes HCC and CCA), the underlying mechanisms of tumorigenesis remain debated. The prevalent view is that tumorigenesis results from an increased mutational burden[8–18]. In line with this view, Brunner et al.[15] showed that cirrhotic liver cells from NASH patients exhibited an increase in mutational load, even though this increase was small and variance between patients was high. In contrast with these observations, we did not observe an increase in mutation rate in individual stem cells of precancerous livers. Additionally, while previous studies have associated specific mutational signatures and gene mutations in HCC to alcohol consumption[44,45], we did not observe an altered mutational landscape in our alcoholic cirrhosis ICOs.

There are several possible explanations for the unchanged mutational landscape in our liver disease stem cells. Firstly, it could be that the cholangiocytes that give rise to the ICO cultures are not the precursors to (pre-)cancerous liver cells. Hepatocellular

carcinoma is derived from hepatocytes and not from cholangiocytes and the cholangiocytes that give rise to ICOs may not be representative for the cholangiocytes that have the potential to develop into cholangiocarcinoma. Secondly, there could be selection for the most stable ASCs in culture, which may not necessarily be the precancerous cells. Thirdly, it may be possible that increased mutagenesis affected only a small proportion of the liver cells and that we have sequenced too few cells to identify the hypermutated cells. Lastly, it may be that our patient cohorts were too small to detect changes in mutational landscape, which may indeed be the case for detecting enrichment of driver gene mutations. However, we could determine via a power analysis (Supplementary Note 1) that the sample sizes in our study were sufficient to detect changes in mutational load and mutational signature contribution similar to other studies which also used tissue derived organoids to investigate mutation accumulation[26,46,47]. It may be however possible to pick up more subtle mutational impacts by increasing sample sizes.

On the other hand, the stem cells that we studied have been exposed to the same environmental stressors as the cells that grew out as tumors (since both have been in contact with the blood stream), so when there would have been a direct mutational impact, we would expect this to be detectable in the cells that we studied. In line with our findings, Brunner et al.[15] also found that alcoholic cirrhosis and NASH livers did not exhibit different mutational signatures in comparison to healthy livers. While the mutational impact of NASH and PSC have not been investigated besides in the Brunner et al.[15] study, alcohol exposure has been shown to lead to DNA damage in vitro[10,13,48] and in blood cells in mice[49], but these studies do not accurately reflect the in vivo liver environmental conditions. It is possible that the rate of cellular endocytosis and/or diffusion is slower in the liver, resulting in less exposure to alcohol than would occur in vitro.

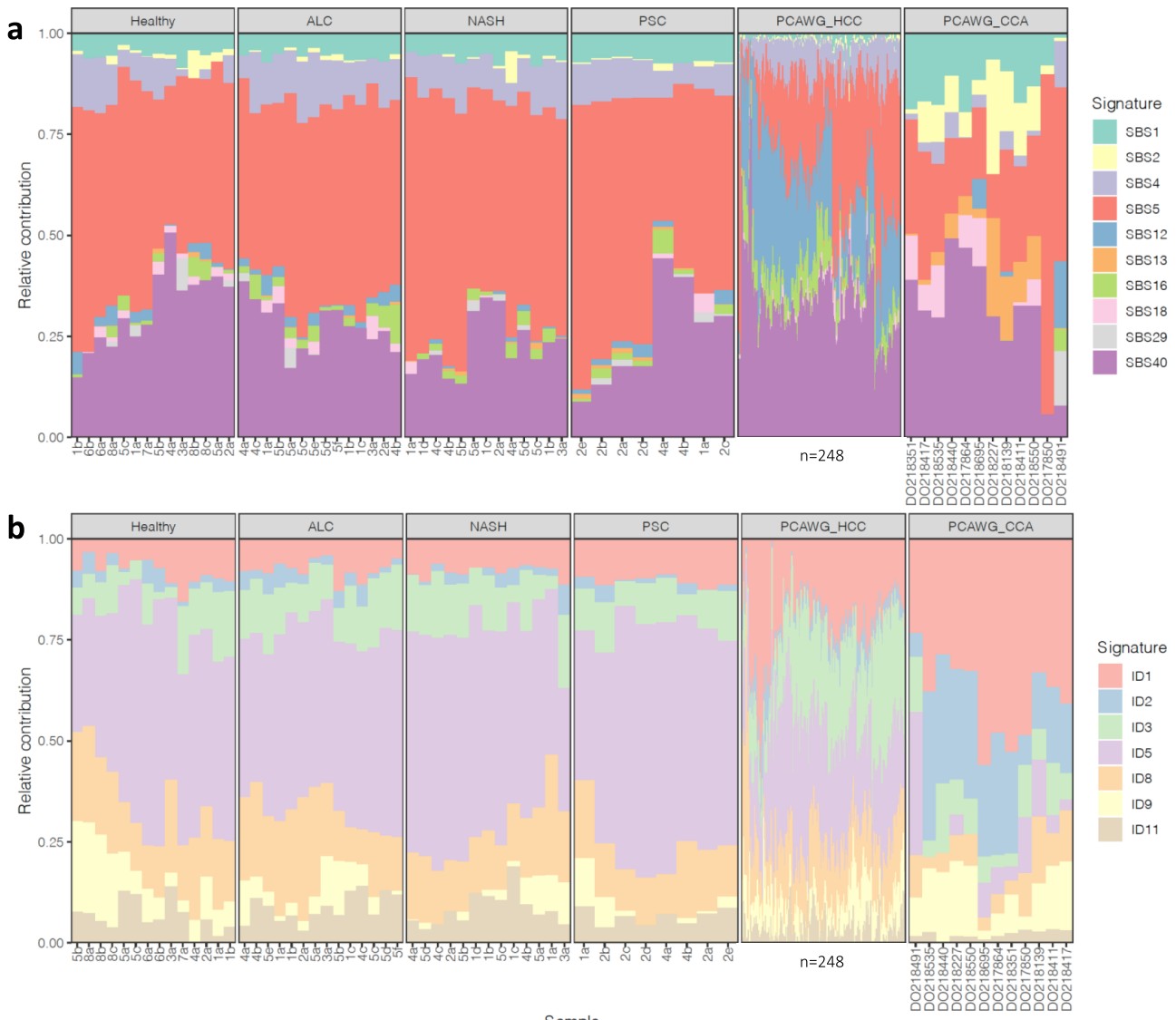

**Fig. 3 Relative contributions of mutational signatures in organoids derived from biopsies of healthy, diseased and cancerous livers. a** Single base substitution (SBS) signatures. **b** Indel (ID) signatures. ALC alcoholic cirrhosis, NASH non-alcoholic steatohepatitis, PSC primary sclerosing cholangitis, HCC hepatocellular carcinoma, CCA cholangiocarcinoma. Profiles for HCC (PCAWG_HCC; $n = 248$) and CCA (PCAWG_CCA; $n = 12$) samples from the Pan-Cancer Analysis of Whole Genomes (PCAWG) consortium are also shown. Hierarchical clustering of samples was performed separately for SBS and ID signatures. Sample names for PCAWG_HCC samples are hidden due to the large number of samples.

Additionally, the aforementioned studies were performed on non-quiescent cells which may rely more on replicative repair, whereas liver cells are generally quiescent and likely rely on non-replicative repair which is faster at repairing alcohol induced DNA damage than replicative repair[8]. Alternatively, liver cells that acquire alcohol induced DNA damage may undergo apoptosis and be replaced by new cells as a result of liver regeneration[50], and these cells in turn lack the mutation footprint caused by the alcohol. Nevertheless, the absence of increased mutational burden in our disease ICOs may suggest that increased mutagenesis is not the primary contributing factor towards tumorigenesis.

Opposed to the view that tumorigenesis arises from mutagenesis, an alternative hypothesis proposes that chronic liver inflammation and cirrhosis (which commonly precedes primary liver cancer[51]) leads to cell death in the liver, requiring normally quiescent liver adult stem cells to proliferate at a much higher rate to regenerate the damaged liver. As a consequence, cells would accumulate more mutations, especially those caused by background mutational

mechanisms related to cell proliferation (e.g., aging-associated mutational signatures). Inflammatory disease conditions would thus provide a "fertile ground" for cells with random and potentially pre-existing (oncogenic) mutations that confer a selective growth advantage to clonally expand[52–54]. Such a phenomenon has been described in mouse models, whereby pancreatic cells within mice with both a pathogenic *Kras* mutation and pancreatitis transitioned into an epigenetic state similar to pancreatic ductal adenocarcinoma, while pancreatic cells in mice with only one or the other retained their original epigenetic state[55]. Additionally, Hepatitis C Virus (HCV)-induced cirrhotic livers showed an increase in the number and size of clonal patches with mutations in genes that are frequently mutated in HCC[50].

Taken together, our findings suggest that mechanisms other than direct mutagenesis drive the transition from healthy to precancerous liver, and highlights the need to explore other potential hypotheses of liver tumorigenesis, including but not limited to the 'fertile ground' hypothesis.

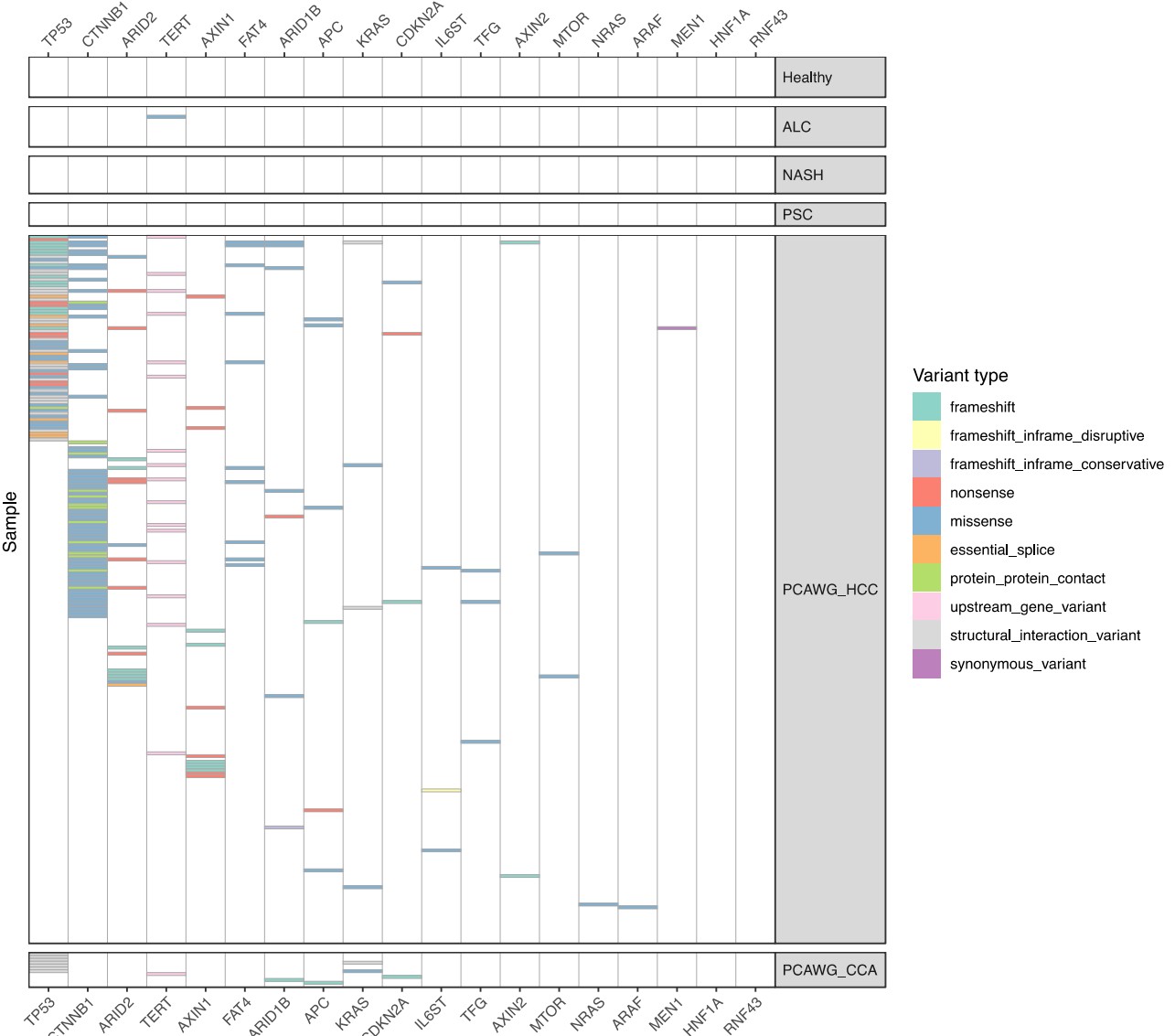

**Fig. 4 Non-synonymous mutations in organoids derived from biopsies of healthy, diseased and cancerous livers.** ALC alcoholic cirrhosis, NASH non-alcoholic steatohepatitis, PSC primary sclerosing cholangitis, HCC hepatocellular carcinoma, CCA cholangiocarcinoma. Profiles for HCC (PCAWG_HCC; $n = 248$) and CCA (PCAWG_CCA; $n = 12$) samples from the Pan-Cancer Analysis of Whole Genomes (PCAWG) consortium are also shown.

## Methods

**Human tissue material**. All human tissue biopsies were obtained in the Erasmus MC—University Medical Center Rotterdam. Liver biopsies from healthy liver donors and patients with alcoholic cirrhosis, nonalcoholic steatohepatitis (NASH) or primary sclerosing cholangitis (PSC) were obtained during liver transplantation procedures. All patients were negative for viral infection and metabolic diseases. The biopsies were collected in cold organ preservation fluid (Belzer UW Cold Storage Solution, Bridge to Life, London, UK) and transported and stored at 4 °C until use. The acquisition of these liver biopsies for research purposes was approved by the Medical Ethical Committee of the Erasmus Medical Center (MEC-2014-060 and MEC-2013-143). Informed consent was provided by all patients involved.

**Generating clonal intrahepatic cholangiocyte organoid cultures from human liver biopsies**. Healthy and diseased liver tissue biopsies were washed in cold DMEM (ThermoFisher) supplemented with 1% fetal calf serum (FCS) and 1% penicillin-streptomycin (wash solution). Subsequently, the tissue was transferred to a petri dish and thoroughly minced with scalpel blades. The minced tissue was transferred to 4 ml digestion solution consisting of EBSS with $Ca^{2+}/Mg^{2+}$ (ThermoFisher) with 1 mg/ml Collagenase type IA (Sigma, C9891) and 0.1 mg/ml DNAse I (Sigma DN25). The tissue was incubated for 30 min at 37 °C with regular shaking. Next, the suspension was passed through a pipet to further break up the tissue and passed through a 70 μm Nylon cell strainer. The cells were washed once with wash solution, followed by two washes in Advanced DMEM F12 supplemented with 1% penicillin–streptomycin, 10 mM HEPES, and 1× Glutamax (all from ThermoFisher). After the final wash, the

cell pellet was resuspended in Matrigel (Corning) and plated in 40 μl droplets per well in prewarmed non-adhesive 24-well plates. The plates were placed at 37 °C in a humidified atmosphere and 5% $CO_2$. After Matrigel had solidified, 500 μl liver organoid establishment medium was added to the wells. Establishment medium consisted of Advanced DMEM F12 supplemented with 1% penicillin–streptomycin, 10 mM HEPES, 1× Glutamax, 10% R-Spondin conditioned medium (produced in house), B27 supplement without Vitamin A (ThermoFisher), N2 supplement (ThermoFisher), 10 mM Nicotinamide (Sigma Aldrich), 1.25 mM N-acetylcysteine (Sigma Aldrich), Primocin, 5 μM A83-01 (Tocris Bioscience), 10 μM Forskolin (Tocris Bioscience), 100 ng/ml FGF-10 (Peprotech), recombinant human Noggin (Peprotech), 10 μM Rho kinase inhibitor (Abmole), hES cell cloning & recovery supplement (Stemgent), 25 ng/ml HGF (Peprotech), 10 nM Gastrin (Tocris), 50 ng/ml human EGF (Peprotech), and 0.3 nM Wnt-surrogate Fc protein (U-protein Express BV). After 2–3 days after isolation, the first intrahepatic cholangiocyte organoids started to appear and establishment medium was switched to maintenance medium consisting of Advanced DMEM F12 supplemented with 1% penicillin-streptomycin, 10 mM HEPES, 1× Glutamax, 10% R-Spondin conditioned medium, B27 supplement without Vitamin A, N2 supplement 10 mM Nicotinamide, 1.25 mM N-acetylcysteine, Primocin, 5 μM A83-01, 10 μM Forskolin, 100 ng/ml FGF-10, 25 ng/ml HGF (Peprotech), 10 nM Gastrin (Tocris), and 50 ng/ml human EGF (Peprotech). The cultures were maintained for 10–14 days after isolation, to enrich for adult stem cells. Subsequently, clonal organoid cultures were generated from these organoid cultures by FACS or by manual selection and expansion of individual organoids[56]. The organoid cultures were further expanded until there was enough material for DNA

isolation. DNA was isolated from all organoid cultures, blood samples, and tissue biopsies using the Qiasymphony (Qiagen). Whole-genome sequencing libraries were generated from 200 ng of genomic DNA according to standard protocols (Illumina). The organoid cultures and control samples were sequenced paired-end ($2 \times 100$ bp) to a depth of at least 30× coverage on the Illumina HiSeq Xten. The hepatocellular carcinoma biopsies were sequenced paired-end ($2 \times 100$ bp) to a depth of at least 60× coverage on the Illumina HiSeq Xten. Whole-genome sequencing was performed at the Hartwig Medical Foundation in Amsterdam, the Netherlands.

**Variant calling**. Germline and somatic variant calling for all samples was performed using the HMF pipeline (https://github.com/hartwigmedical/pipeline; v4.8)[57]. Briefly, reads were mapped to GRCh37 using BWA-MEM v0.7.5a with duplicates being marked for filtering. Indels were realigned using GATK v3.4.46 IndelRealigner. GATK Haplotype Caller v3.4.46 was used for calling germline variants in the reference sample. For somatic SNV and indel variant calling, GATK BQSR3 was first used to recalibrate base qualities, followed by Strelka v1.0.14 for the variant calling itself. Somatic structural variant calling was performed using GRIDSS v1.8.0. Copy-number calling was performed using PURity & PLoidy Estimator (PURPLE), that combines B-allele frequency (BAF), read depth, and structural variants to estimate the purity and copy number profile of a tumor sample. For SNVs and indels, downstream analyses were performed only on variants marked as "PASS".

**Mutation context analysis**. The counts of single base substitution (SBS), double base substitution (DBS), indel and structural (SV) variant contexts were determined from somatic VCF files using the R package *mutSigExtractor* (https://github.com/UMCUGenetics/mutSigExtractor; v1.23). The mutation contexts of all mutation types are described in COSMIC (https://cancer.sanger.ac.uk/cosmic/signatures), except for the SV contexts. The 16 SV contexts were composed of the SV type (deletion, duplication, inversion, translocation) and the SV length (1–10 kb, 10–100 kb, 100 kb–1 Mb, 1–10 Mb, and >10 Mb). Note that SV length is not applicable for translocations. The mutation context spectra for each sample group are shown in Supplementary Figs. 4–7.

To perform mutational signature analysis, we selected the SBS and DBS signatures that were present in at least 10% of liver cancer (Liver-HCC) or biliary cancer (Biliary-AdenoCA) PCAWG samples (https://dcc.icgc.org/releases/PCAWG/mutational_signatures/Signatures_in_Samples/SP_Signatures_in_Samples)[20]. We then fitted the SBS and DBS mutation contexts to these selected signatures using the fitToSignatures() function from mutSigExtractor (which employs the non-negative least-squares method) to obtain absolute signature contributions. Relative signature contribution per sample was calculated by dividing the absolute contributions by the total signature contribution.

Mutation context and signature absolute contributions per sample can be found in Supplementary Data 3.

**Selection of liver and biliary cancer driver genes**. A catalog of driver genes by cancer type was downloaded from Intogen (https://www.intogen.org/download; release 2020.02.01). From the Compendium_Cancer_Genes.tsv file, we selected genes where CANCER_TYPE was "HC" or "CH" (hepatocellular carcinoma and cholangiocarcinoma, respectively), and CGC_CANCER_GENE was "TRUE". Additionally, *TERT* has been reported by the PCAWG consortium[43] as a known HCC driver gene and was thus also included.

**Identifying non-synonymous mutations**. The mutation type of each somatic SNV/indel was determined by SnpEff (http://snpeff.sourceforge.net/; v4.3t). The following variant types were considered non-synonymous mutations: out-of-frame frameshifts, disruptive inframe frameshifts, nonsense, missense, splice variants. We also considered a mutation as non-synonymous if it was annotated as VUS, likely pathogenic, or pathogenic by ClinVar (https://www.ncbi.nlm.nih.gov/clinvar/; GRCh37, database date 2020-02-24), or if it was a hotspot mutation. The underlying code for annotating non-synonymous mutations can be found at https://github.com/UMCUGenetics/geneDriverAnnotator (v1.0).

The dndscv R package[52] was used to identify genes that were enriched for non-synonymous mutations. Briefly, this package computes the (local) background mutation rates and sequence composition of genes to calculate the background mutation rate for each gene. A likelihood ratio test is subsequently performed to identify genes that are significantly hit by nonsynonymous mutations. dndscv was run separately for each disease status group (i.e., separately for healthy ICOs, separately for NASH ICOs, etc) using all the somatic mutations from the respective group.

**Statistics and reproducibility**. All statistical analyses were performed in R (v4.0.3). To correlate the number of mutations with the age of each patient from which each biopsy was derived (as shown in Fig. 2 and Supplementary Fig. 1), we first assessed normality of the mutational load per mutation type per disease status group was using the Shapiro test (shapiro.test() function) (Supplementary Table 1). This confirmed that the mutational load was normally distributed ($p > 0.05$), with near normality for SBS load in PSC samples ($p = 0.03$) and indel load in NASH samples ($p = 0.05$). Then, the lme() function from the nlme (v3.1) package was used to fit a linear mixed effects regression, with 95% confidence intervals being calculated using the intervals() function from the nlme package. Here, "patient"

was modelled as a random effect to account for having different numbers of organoids per patient. Additionally, the intercept was fixed to zero as it was assumed that a patient has no somatic mutations at birth. A two-sided $Z$-test was used to calculate the difference between two regressions. The $Z$-statistic was first calculated using the slope ($m$) and standard errors (SE) of the two regressions (Eq. 1), which was then used to calculate a $p$-value using the pnorm() function (Eq. 2). A one-sided $F$-test was performed to calculate whether the variance of the regression of diseased ICOs was greater than that of the healthy ICOs. The $F$-statistic was first calculated by dividing the variance of the two regressions (extracted from the output of the lme() function) (Eq. 3), which was then used to calculate a $p$-value using the pf() function (Eq. 4).

$$Z = \frac{m_1 - m_2}{\sqrt{\mathrm{SE}_1^2 - \mathrm{SE}_2^2}} \tag{1}$$

$$p = 2 \times \mathrm{pnorm}(-|Z|) \tag{2}$$

$$F = \mathrm{var}_{\mathrm{disease}} / \mathrm{var}_{\mathrm{healthy}} \tag{3}$$

$$p = 1 - pf(F) \tag{4}$$

To determine whether there was a significant increase in mutation context load in disease versus healthy ICOs (as shown in Supplementary Figs. 4–7), Wilcoxon rank sum tests (using the wilcox.test() function) were performed per mutation context. Bonferroni multiple testing correction was then applied to the resulting $p$-values (using the p.adjust() function).

**Reporting summary**. Further information on research design is available in the Nature Research Reporting Summary linked to this article.

## Data availability

The BAM files from the whole-genome sequencing data generated in the current study are available at EGA (https://www.ebi.ac.uk/ega/home) under accession numbers EGAS00001002983 and EGAS00001005384. BAM files from hepatocellular carcinoma and cholangiocarcinoma patients from the Pan-Cancer Analysis Whole Genomes (PCAWG) consortium were obtained under request number DACO-5333. For access to the PCAWG BAM files, researchers will need to request access via the ICGC Data Access Compliance Office (DACO; https://daco.icgc.org/). The VCF and tabular files produced from somatic variant calling are available at https://zenodo.org/record/556238[58].

## Code availability

The code for the Hartwig Medical Foundation (HMF) germline and somatic variant calling pipeline is available at https://github.com/hartwigmedical/pipeline. The code used for data processing and generating the figures is available at https://github.com/UMCUGenetics/Diseased_livers[59].

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

## Acknowledgements

This study was financially supported by the research program InnoSysTox (project number 114027003), by the Netherlands Organisation for Health Research and Development (ZonMw), by the Dutch Cancer Society (project number 10496) and is part of the Oncode Institute, which is partly financed by the Dutch Cancer Society and was funded by the gravitation program CancerGenomiCs.nl from the Netherlands Organisation for Scientific Research (NWO). The authors would like to thank Hartwig Medical Foundation and the Utrecht Sequencing Facility for performing the whole genome sequencing. The Utrecht Sequencing Facility is subsidized by the University Medical Center Utrecht, Hubrecht Institute, and Utrecht University.

## Author contributions

R.L., J.J., J.I., M.D., and M.V. collected liver biopsies. M.J., E.K., and N.B. performed organoid culturing. L.N., M.J., M.L., B.R., R.J., and S.B. performed bioinformatic analyses. M.J., E.K., M.V., R.B., L.L., and E.C. were involved in the conceptual design of this study. L.N., E.K., M.J., and E.C. wrote the manuscript. All authors provided textual comments and have approved the manuscript. E.K., R.B., L.L., and E.C. supervised this study.

## Competing interests

The authors declare no competing interests.
