## [Transparent Peer Review File · Communications Biology]

Reviewers' comments:

Reviewer #1 (Remarks to the Author):

In this work, Nguyen, Jager and colleagues use clonal amplification of cholangiocyte progenitors to determine mutation burden and pattern in human liver with various disease states. Given that alcoholic and non-alcoholic fatty liver disease are risk factors of hepatocellular carcinoma (HCC), it is hypothesized that this may in part be due to increased mutation in liver somatic tissue. Contrary to the expectation, the authors find that cholangiocyte stem cells do not differ significantly in mutation burden and pattern compared to cells from healthy controls, and it is concluded that increased mutation is not the main mechanism by which these disease states contribute to liver cancer. The work combines state-of-the-art techniques like organoid culture and bioinformatic analysis of WGS data, and the experimental design is robust (with several patients and samples/patient). This work is complementary to recent studies using liver microdissections (Brunner et al, Nature 2019) and has important findings for our understanding of the etiology of liver cancer. I would like to recommend this work for publication in Communications Biology after the following points have been addressed:

1. My main concern is that I believe that the relationship between 'liver cancer', HCC, CGA and their respective 'cells of origin' (hepatocytes and cholangiocytes) needs to be clarified and spelt out more clearly. Only in the discussion do we read 'HCC is derived from hepatocytes and not from cholangiocytes' when I feel this information should be presented earlier.

The authors use cholangiocyte progenitors to assess mutation in liver tissue because these cells can be readily amplified in vitro into clonal cultures, to provide enough DNA for sequencing. Although human fetal hepatocyte organoids have been reported recently, this is not yet possible from hepatocytes from adult liver. Therefore, mutation in cholangiocytes are used to draw conclusions about mechanisms of mutagenesis in hepatocytes, HCC and 'liver cancer'. I feel at times these lines are blurred, confusing/misleading the reader. For example, in the introduction lines 38-40: 'Several factors have been linked to increased liver cancer risk including chronic alcohol consumption, MAFLD, NASH, which can be caused by (...)'. References 3 and 4 refer to HCC specifically. Are MAFLD and NASH also predisposing to CGA? If so, these references should be added here. If not, 'liver cancer' should be replaced by HCC. Similarly, I think 'liver' in the title should be replaced by cholangiocyte or liver cholangiocyte stem cells.

SBS12 and SBS16 appear to be HCC-specific processes (as seen in Figure 3a). Although Brunner 2019 argues against this, future and similar studies to the one presented here but carried out in hepatocytes might reveal that these diseases do contribute to mutation in these cells.

2. More detail is needed on the clinical history of the patients or clinical diagnosis of alcohol steatohepatitis, NASH and PSC, see for example methods in Brunner 2019. I believe this is relevant to get an idea of how long these cells have been exposed to the 'mutagenic' insult. If, for example, a 60-year-old patient has only abused alcohol for the last three years, then it may be hard to detect mutations arising during this period from the cumulative mutations over the first 57 years.

3. Lines 225-227: "Despite the association between liver disease and primary liver cancer, the underlying mechanisms of tumorigenesis remain debated. The prevalent view is that tumorigenesis results from an increased mutational burden". The authors should direct the readers to papers/reviews with the 'prevalent view' by citing relevant literature.

4. A unique aspect of mutagenesis in the liver is the presence of SBS16, or 'transcription-coupled damage', which is also induced by alcohol consumption (Letouze 2017). An increase in genome-wide SBS16 was not detected in cholangiocyte stem cells, this may well be because this process is more prevalent in HCC/hepatocytes. However, the presence of SBS16 is more obvious in very highly transcribed genes (Letouze 2017, also Manticorena Science 2018). If the mutational burden/load allows, the authors have a unique opportunity to provide more insight into this process by comparing mutations in highly transcribed genes between Healthy, NASH and PSC vs ALC samples.

5. I think the use of 'Precancerous' (in the title or manuscript) is misleading. Alcohol, NASH and PSC predispose to liver cancer but there is no way of telling if the patients whose samples were analyzed would go on to develop cancer. What fraction of people that drink, with NASH or PSC develop liver cancer? This is equivalent to saying that anyone who smokes has precancerous tissue.

Reviewer #2 (Remarks to the Author):

In this study, the authors tried to uncover the association between liver disease and primary liver cancer in term of mutational signatures and suggested environmental conditions rather than direct mutagenesis drive the transition from healthy to precancerous liver. The views put forward in this manuscript is appealing, that precancerous liver disease conditions do not result in a detectable increased accumulation of mutations, nor altered mutation types in individual liver stem cells, which is different from traditional cognition. However, the precise of the modelling system of ICOs is doubtful. Furthermore, it may need more evidence to get rid of interference for the negative outcome, then the conclusion could be reliable.

1. As illustrated by other researchers and also mentioned by the authors "Hepatocellular carcinoma is derived from hepatocytes and not from cholangiocytes and the cholangiocytes that give rise to ICOs may not be representative for the cholangiocytes that have the potential to develop into cholangiocarcinoma." Therefore, do the authors not think that the using of ICOs system in this study for analysis lacks feasibility basis?

2. "The cultures were maintained for 10-14 days after isolation, to enrich for adult stem cells. Subsequently, clonal organoid cultures were generated from these organoid.....", "The organoid cultures were further expanded until there was enough material for DNA isolation"— It's well know that the long-term in vitro culture may alter the biological features of cells or even mutational signatures, the authors should confirm the cultured ICOs maintained the mutational similarity to that before isolation.

3. Mutations are known as time related, the duration of these disease in this passage should be mentioned. What is more, a comparison of mutations among early, middle and advanced stage of these disease may get the illustration clearer.

4. The sample size of these precancerous liver diseases seems relatively small numbers, only 5 ALC patients, 5 NASH patients, 3 PSC patients. For HCC_multibiopsy group, only one patient was involved. From figure 4, we are informed that even the HCC_multibiopsy samples, have no mutation, which have a big difference with PCAWG_HCC. Its contingency cannot be rule out from only 5 samples, in addition, reliability of test reports may be doubted.

5. Although in this study, the mentioned diseases seemed not directly resulted in increased mutagenesis in liver stem cells, the indirect effects from live diseases which involves changed environment condition could not be excluded. So please modify the title for perfect summarization

6. "We ultimately quantified the presence of 10 SBS and 7 indel signatures in our ICOs as well as in the PCAWG HCC and CCA samples." —data not shown.

7. In the figure 1, samples in 1A seems confusion, readers cannot fully understand the exact meaning of those number.

8. In the figure 2, different number represent different samples with the same color, which cannot be distinguished in the picture, and why there are different mutations in one sample at one age point?

Reviewer #3 (Remarks to the Author):

The manuscript „Precancerous alcoholic steatohepatitis, NASH and PSC disease conditions do not result in increased mutagenesis in liver stem cells" from Nguyen and colleagues uses whole-genome

sequencing to analyze the accumulation of mutations in intrahepatic cholangiocyte organoids (ICOs) that were cultivated from clonally expanded single liver stem cells.

This is a very interesting approach for the detection of early mutations which may occur already in the premalignant liver of patients with chronic liver disease and which may lead to tumor formation.

Why do the authors focus on patients with alcoholic steatohepatitis, non-alcoholic steatohepatitis and primary sclerosing cholangitis? Why did they not include patients with viral infections?

What was the rationality to use intrahepatic cholangiocyte organoids (ICOs) instead of hepatocyte-derived organoids for this study? HCCs would be derived primarily from transformed hepatocytes and not from transformed cholangiocytes.

The lack of driver gene mutations in ICOs with the exception of one mutation in the TERT gene is not surprising. To detect driver gene mutations in patients with chronic liver disease a much higher number of organoids must be generated and analyzed. The limitations for the detection of driver gene mutations by using ICOs should be discussed. I would not conclude that those genes are not necessarily required for HCC development, due to the low number of sample size (HCC and ICOs).

Reviewers' comments:

Reviewer #1 (Remarks to the Author):

In this work, Nguyen, Jager and colleagues use clonal amplification of cholangiocyte progenitors to determine mutation burden and pattern in human liver with various disease states. Given that alcoholic and non-alcoholic fatty liver disease are risk factors of hepatocellular carcinoma (HCC), it is hypothesized that this may in part be due to increased mutation in liver somatic tissue. Contrary to the expectation, the authors find that cholangiocyte stem cells do not differ significantly in mutation burden and pattern compared to cells from healthy controls, and it is concluded that increased mutation is not the main mechanism by which these disease states contribute to liver cancer. The work combines state-of-the-art techniques like organoid culture and bioinformatic analysis of WGS data, and the experimental design is robust (with several patients and samples/patient). This work is complementary to recent studies using liver microdissections (Brunner et al, Nature 2019) and has important findings for our understanding of the etiology of liver cancer. I would like to recommend this work for publication in Communications Biology after the following points have been addressed:

1. My main concern is that I believe that the relationship between 'liver cancer', HCC, CGA and their respective 'cells of origin' (hepatocytes and cholangiocytes) needs to be clarified and spelt out more clearly. Only in the discussion do we read 'HCC is derived from hepatocytes and not from cholangiocytes' when I feel this information should be presented earlier.

We have indicated in the introduction that hepatocellular carcinoma originates from hepatocytes and cholangiocarcinoma from cholangiocytes.

The authors use cholangiocyte progenitors to assess mutation in liver tissue because these cells can be readily amplified in vitro into clonal cultures, to provide enough DNA for sequencing. Although human fetal hepatocyte organoids have been reported recently, this is not yet possible from hepatocytes from adult liver. Therefore, mutation in cholangiocytes are used to draw conclusions about mechanisms of mutagenesis in hepatocytes, HCC and 'liver cancer'. I feel at times these lines are blurred, confusing/misleading the reader. For example, in the introduction lines 38-40: 'Several factors have been linked to increased liver cancer risk including chronic alcohol consumption, MAFLD, NASH, which can be caused by (...)'. References 3 and 4 refer to HCC specifically. Are MAFLD and NASH also predisposing to CGA? If so, these references should be added here. If not, 'liver cancer' should be replaced by HCC. Similarly, I think 'liver' in the title should be replaced by cholangiocyte or liver cholangiocyte stem cells.

MAFLD and NASH confer risk for intrahepatic CCA. We have added a reference (Wongjarupong et al 2017) for this in the introduction.

We have also replaced 'liver cancer' with 'HCC' and/or 'CCA' when this was ambiguous. Additionally, we have changed the title to "Liver diseases that predispose patients to liver cancer do not result in increased mutagenesis in liver cholangiocyte stem cells".

SBS12 and SBS16 appear to be HCC-specific processes (as seen in Figure 3a). Although Brunner 2019 argues against this, future and similar studies to the one presented here but carried out in hepatocytes might reveal that these diseases do contribute to mutation in these cells.

We indeed agree with the reviewer here.

2. More detail is needed on the clinical history of the patients or clinical diagnosis of alcohol steatohepatitis, NASH and PSC, see for example methods in Brunner 2019. I believe this is relevant to get an idea of how long these cells have been exposed to the ‘mutagenic’ insult. If, for example, a 60-year-old patient has only abused alcohol for the last three years, then it may be hard to detect mutations arising during this period from the cumulative mutations over the first 57 years.

We agree with the reviewer that this important information needs to be included in the manuscript. We have now included this additional clinical information in Supplementary data 1. In general, information on the number of years of alcohol abuse is not very reliable because most patients have a tendency to underestimate the length of this period. For our study, this information is, unfortunately, only available for 2 patients, namely patients ALC2 (3 years) and ALC5 (10 years). Both patients had similar mutational loads, with ALC2 having ~2300 SBSs and ~550 indels, and ALC5 having ~3000 SBSs and ~600 indels (see excerpt from Figure 2 below). Thus even after a long exposure to alcohol abuse (for patient ALC5), no clear mutational impact can be detected.

3. Lines 225-227: “Despite the association between liver disease and primary liver cancer, the underlying mechanisms of tumorigenesis remain debated. The prevalent view is that tumorigenesis results from an increased mutational burden”. The authors should direct the readers to papers/reviews with the ‘prevalent view’ by citing relevant literature.

References have been added to this sentence.

4. A unique aspect of mutagenesis in the liver is the presence of SBS16, or ‘transcription-coupled damage’, which is also induced by alcohol consumption (Letouze 2017). An increase in genome-wide SBS16 was not detected in cholangiocyte stem cells, this may well be because this process is more prevalent in HCC/hepatocytes. However, the presence of SBS16 is more obvious in very highly transcribed genes (Letouze 2017, also Manticorena Science 2018). If the mutational burden/load allows, the authors have a unique opportunity to provide more insight into this process by comparing mutations in highly transcribed genes between Healthy, NASH and PSC vs ALC samples.

This is an interesting suggestion, but unfortunately there are too few mutations to perform this analysis. The contribution of SBS12 and SBS16 is only ~5% in NASH/PSC/ALC samples (Figure 3a), and the average SBS load per sample is ~3000 mutations (Figure 2). The number of SBS12 and SBS16 mutations in total is thus ~150 per sample. Further filtering for SBS12 and SBS16 mutations within highly transcribed genes would thus not yield enough mutations for the suggested analysis.

5. I think the use of ‘Precancerous’ (in the title or manuscript) is misleading. Alcohol, NASH and PSC predispose to liver cancer but there is no way of telling if the patients whose samples were analyzed would go on to develop cancer. What fraction of people that drink, with NASH or PSC develop liver cancer? This is equivalent to saying that anyone who smokes has precancerous tissue.

We have changed the title to “Liver diseases that predispose patients to liver cancer do not result in increased mutagenesis in liver cholangiocyte stem cells”.

Reviewer #2 (Remarks to the Author):

In this study, the authors tried to uncover the association between liver disease and primary liver cancer in term of mutational signatures and suggested environmental conditions rather than direct mutagenesis drive the transition from healthy to precancerous liver. The views put forward in this manuscript is appealing, that precancerous liver disease conditions do not result in a detectable increased accumulation of mutations, nor altered mutation types in individual liver stem cells, which is different from traditional cognition. However, the precise of the modelling system of ICOs is doubtful. Furthermore, it may need more evidence to get rid of interference for the negative outcome, then the conclusion could be reliable.

1. As illustrated by other researchers and also mentioned by the authors “Hepatocellular carcinoma is derived from hepatocytes and not from cholangiocytes and the cholangiocytes that give rise to ICOs may not be representative for the cholangiocytes that have the potential to develop into cholangiocarcinoma.” Therefore, do the authors not think that the using of ICOs system in this study for analysis lacks feasibility basis?

We reasoned that cholangiocytes are also suitable for the study of somatic mutation accumulation as a result of the diseased liver environment, because cholangiocytes are exposed to the same environmental conditions as the other liver cell types (Sia et al 2016). This text has also been added to the beginning of the results section.

2. “The cultures were maintained for 10-14 days after isolation, to enrich for adult stem cells. Subsequently, clonal organoid cultures were generated from these organoid.....”, “The organoid cultures were further expanded until there was enough material for DNA isolation”— It’s well know that the long-term in vitro culture may alter the biological features of cells or even mutational signatures, the authors should confirm the cultured ICOs maintained the mutational similarity to that before isolation.

Ours is a well-verified method to define in vivo acquired mutations in individual stem cells (Blokzijl et al 2016). We have also shown that, in vitro, human liver stem cell derived organoids acquire about 7.2 SBSs per genome per population doubling (Kuijk et al. 2020). Since we have cultured our ICOs for about 14 days, we expect a total of ~53 SBS being acquired in vitro. This is negligible in comparison to the ~3000 SBSs on average (Figure 2) that each sample had acquired prior to in vitro culture.

3. Mutations are known as time related, the duration of these disease in this passage should be mentioned. What is more, a comparison of mutations among early, middle and advanced stage of these disease may get the illustration clearer.

We looked at the advanced stage of liver diseases, where the patients needed to undergo liver transplantation. If these diseases induce mutations, we would anticipate to find them in these patients. We did not find mutations induced by these advanced stage diseases. It is therefore highly unlikely to assume to find more mutations in early and middle stages of these diseases.

4. The sample size of these precancerous liver diseases seems relatively small numbers, only 5 ALC patients, 5 NASH patients, 3 PSC patients. For HCC_multibiopsy group, only one patient was involved. From figure 4, we are informed that even the HCC_multibiopsy samples, have no mutation, which have a big difference with PCAWG_HCC. Its contingency cannot be rule out from only 5 samples, in addition, reliability of test reports may be doubted.

It is very difficult to acquire tissue from diseased and healthy liver patients. The current study was conducted over a period of five years, mainly as a result of limited availability of liver biopsies. It is therefore not feasible for us to significantly increase sample sizes. What is clear from our study is that there are no clear mutational signals, though it might be possible to pick up more subtle mutational impacts by increasing sample sizes.

However, we could determine via a power analysis that the sample sizes in our study were sufficient to detect large changes in mutational load and mutational signature contribution similar to other studies which also used tissue derived organoids to investigate mutation accumulation (Jager et al 2019, Drost et al 2017, Kuijk et al 2020). This power analysis has been included as a section in the Supplementary notes and mentioned in the discussion.

Since only one patient was involved for the HCC_multibiopsy group, we agree with the reviewer and have removed the results and text relating to this group from the manuscript.

5. Although in this study, the mentioned diseases seemed not directly resulted in increased mutagenesis in liver stem cells, the indirect effects from live diseases which involves changed environment condition could not be excluded. So please modify the title for perfect summarization

Our method captures mutations that are the result of all effects including indirect effects. The title has now been changed to: “Liver diseases that predispose patients to liver cancer do not result in increased mutagenesis in liver cholangiocyte stem cells”

6. “We ultimately quantified the presence of 10 SBS and 7 indel signatures in our ICOs as well as in the PCAWG HCC and CCA samples.” —data not shown.

A reference to figure 3 has been added to this sentence.

7. In the figure 1, samples in 1A seems confusion, readers cannot fully understand the exact meaning of those number.

We have now named the samples e.g. ALC1a, where the number indicates the patient number and the letter indicates the clone.

8. In the figure 2, different number represent different samples with the same color, which cannot be distinguished in the picture, and why there are different mutations in one sample at one age point?

Each point represents a different clone, where clones from the same patient are at the same age point. We have now labelled the points by patient number/clone letter to make this clearer.

Reviewer #3 (Remarks to the Author):

The manuscript „Precancerous alcoholic steatohepatitis, NASH and PSC disease conditions do not result in increased mutagenesis in liver stem cells” from Nguyen and colleagues uses whole-genome sequencing to analyze the accumulation of mutations in intrahepatic cholangiocyte organoids (ICOs) that were cultivated from clonally expanded single liver stem cells.

This is a very interesting approach for the detection of early mutations with may occur already in the premalignant liver of patients with chronic liver disease and which may lead to tumor formation.

Why do the authors focus on patients with alcoholic steatohepatitis, non-alcoholic steatohepatitis and primary sclerosing cholangitis? Why did they not include patients with viral infections?

We focused on these disease conditions as they are the major risk factors towards liver cancer (Llovet et al 2021, Fung et al 2019). We did not include patients with Hepatitis B virus (HBV) integrations as it has already been demonstrated that HBV integrations can lead to cancer via mutation of cancer driver genes (Llovet et al 2021).

What was the rationality to use intrahepatic cholangiocyte organoids (ICOs) instead of hepatocyte-derived organoids for this study? HCCs would be derived primarily from transformed hepatocytes and not from transformed cholangiocytes.

We have chosen to use ICOs for studying mutational accumulation as cholangiocyte derived organoids can be readily clonally expanded while organoids derived from hepatocytes cannot. Since hepatocytes and cholangiocytes are exposed to the bloodstream and are thus subject to similar environmental conditions, we expect similar mutation accumulation in both cell types. This has now been mentioned at the start of the results section

The lack of driver gene mutations in ICOs with the exception of one mutation in the TERT gene is not surprising. To detect driver gene mutations in patients with chronic liver disease a much higher number of organoids must be generated and analyzed. The limitations for the detection of driver gene mutations by using ICOs should be discussed. I would not conclude that those genes are not necessarily required for HCC development, due to the low number of sample size (HCC and ICOs).

We have acknowledged in the results and discussion sections that our small sample sizes limit our ability to find enriched driver gene mutations in the diseased liver ICOs.

REVIEWERS' COMMENTS:

Reviewer #1 (Remarks to the Author):

The authors have addressed my main points and I therefore recommend the paper for publication.

Reviewer #2 (Remarks to the Author):

The most of the concerns are addressed in this revised version.

The author estimated ~53 SBS being acquired in vitro, which is negligible in comparison to that before culture. They should also provide evidences that the mutations did not lose or be repaired after long-term culture, either data or references is OK.

Reviewer #3 (Remarks to the Author):

The authors have answered all of my questions satisfactorily.

Reviewer comment:

The author estimated ~53 SBS being acquired in vitro, which is negligible in comparison to that before culture. They should also provide evidences that the mutations did not lose or be repaired after long-term culture, either data or references is OK.

Response:

It is not clear to us what the reviewer means by losing mutations or repair of mutations. Maybe the reviewer means that a mutation reverts to its original base, e.g. when a C > A change reverts back to an A > C. Since a mutation is fixed in the genome, "losing" a mutation by undergoing the reverse mutation is highly unlikely, because it would mean that the same nucleotide out of the 6 billion available nucleotides is damaged again followed by faulty DNA repair resulting in the original base (and not any of the other 2 possible bases). Even if this would happen, it is also not clear why that would happen more often in one liver disease condition and not another. The reviewer also asks for evidence that the mutations are not repaired during long-term culture. Mutations are (not damage (unlike DNA damage that precedes the mutation), so there is nothing to be repaired.